# *Coptis chinensis* Franch Directly Inhibits Proteolytic Activation of Kallikrein 5 and Cathelicidin Associated with Rosacea in Epidermal Keratinocytes

**DOI:** 10.3390/molecules25235556

**Published:** 2020-11-26

**Authors:** Kyung-Baeg Roh, De-Hun Ryu, Eunae Cho, Jin Bae Weon, Deokhoon Park, Dae-Hyuk Kweon, Eunsun Jung

**Affiliations:** 1Biospectrum Life Science Institute, Yongin 16827, Korea; biosh@biospectrum.com (K.-B.R.); biosc@biospectrum.com (D.-H.R.); biozr@biospectrum.com (E.C.); biohy@biospectrum.com (J.B.W.); pdh@biospectrum.com (D.P.); 2Department of Integrative Biotechnology, College of Biotechnology and Bioengineering, Sungkyunkwan University, Suwon 16419, Korea; dhkweon@skku.edu

**Keywords:** rosacea, *Coptis chienesis*, kallikrein 5, cathelicidin, 1,25 (OH)_2_VD_3_, *Demodex*

## Abstract

Rosacea is a common and chronic inflammatory skin disease that is characterized by dysfunction of the immune and vascular system. The excessive production and activation of kallikerin 5 (KLK5) and cathelicidin have been implicated in the pathogenesis of rosacea. *Coptis chinensis* Franch (CC) has been used as a medicinal herb in traditional oriental medicine. However, little is known about the efficacy and mechanism of action of CC in rosacea. In this study, we evaluate the effect of CC and its molecular mechanism on rosacea in human epidermal keratinocytes. CC has the capacity to downregulate the expression of KLK5 and cathelicidin, and also inhibits KLK5 protease activity, which leads to reduced processing of inactive cathelicidin into active LL-37. It was determined that CC ameliorates the expression of pro-inflammatory cytokines through the inhibition of LL-37 processing. In addition, it was confirmed that chitin, an exoskeleton of *Demodex* mites, mediates an immune response through TLR2 activation, and CC inhibits TLR2 expression and downstream signal transduction. Furthermore, CC was shown to inhibit the proliferation of human microvascular endothelial cells induced by LL-37, the cause of erythematous rosacea. These results demonstrate that CC improved rosacea by regulating the immune response and angiogenesis, and revealed its mechanism of action, indicating that CC may be a useful therapeutic agent for rosacea.

## 1. Introduction

Rosacea is a chronic inflammatory disease of the facial skin, characterized by erythema, papules, pustules telangiectasia, and edema, or a combination of these symptoms. The pathophysiology of rosacea has yet to be fully elucidated. However, recent studies suggest that rosacea be attributed to the dysregulation of the neurovascular system and the innate immune system. The dysregulation of the innate immune system is supported by an augmentation of the cathelicidin innate immunity pathway [1,2,3,4,5]. Patients with rosacea express upregulated levels of cathelicidin LL-37, a biologically active form of cathelicidin and kallikrein-related peptidase 5 (KLK5), tryptic serine protease expressed in stratum corneum [6]. Since KLK5 is responsible for the cleavage of inactive cathelicidin into active LL-37, it is implicated, in that elevated levels of LL-37 are the result of high levels of KLK5 in the lesional skin of rosacea [2,6]. Cathelicidin LL-37 exhibits multiple functions that influence various processes in immunity, because it has the capacity to kill microbes and potentially to trigger an immune response. It is described as an alarmin that promotes leukocyte chemotaxis, and can induce the activation of macrophages, mast cells, endothelial cells, and keratinocytes in rosacea [7,8,9]. Therefore, an agent that inhibits the expression of KLK5, or blocks its activity, may prevent the expression of LL-37.

A number of factors are known to induce rosacea, which include UV radiation, *Demodex* colonization, microbial stimuli, heat, stress, and genetic predisposition [10]. Among these factors that may contribute to the aggravation of rosacea, UV exposure is probably one of the most important. Recent studies have shown that while low 1,25-(OH)_2_-vitamin D_3_ (VD_3_) levels are observed in inflammatory skin diseases, such as atopic dermatitis, psoriasis, and chronic urticaria, patients with rosacea show relatively high serum VD_3_ levels, compared to healthy controls [11,12]. Therefore, UV radiation as a source of VD_3_ production represents one of the critical triggering factors for rosacea.

*Demodex folliculorum* is a species of mite that is readily found in the pilosebaceous units of human, which shows a marked increase of the density of *Demodex* on the facial skin of rosacea, compared to healthy controls [13,14]. The increased number of *Demodex* mites is likely to induce an inflammatory response in rosacea lesion through activation of the toll-like receptor 2 (TLR2) pathway [15]. Activation of TLR2 induces the expression of inflammatory mediators, IL-1β, IL-8, TNF-α, and the inflammasome and chitin released from *Demodex*, act as a TLR2 ligand, causing calcium influx, which in turn induces KLK5 expression [16].

Standard agents approved by the FDA for inflammatory rosacea include topical metronidazole, oral doxycycline, ivermectin, and azelaic acid, while for the treatment of erythema of rosacea, topical α-adrenergic receptor agonists, such as brimonidine tartrate and oxymetazoline hydrochloride, prove effective. The therapeutic efficacy of doxycycline, metronidazole, ivermectin, and azelaic acid, used as anti-inflammatory agents, is directly or indirectly associated with the inhibition of KLK5 activation [17]. Adrenergic receptor agonists, brimonidine tartrate and oxymetazoline hydrochloride, have an immediate effect on facial erythema in rosacea [17]; however, this is a temporary symptom-relieving effect, which is difficult to effectively apply to chronic inflammatory rosacea. Currently, although studies on the bioactive products derived from natural products targeting KLK5 have been reported [18], these studies have been limited.

*Coptis chinensis* Franch (CC) is a species of flowering plant that is used as a medicinal herb in traditional oriental medicine. Recent studies have demonstrated that CC shows a broad range of biological activities, including anti-carcinogenic, anti-inflammatory, anti-diabetic, neuroprotective, anti-microbial, and anti-cancer [19,20,21,22,23,24]. However, the effects of CC on rosacea and their mechanism of actions have not yet been fully understood.

In this study, we investigate whether the ability of CC to inhibit KLK5 could influence the activation of cathelicidin and induction of inflammatory response and erythema in human epidermal keratinocytes. In addition, we investigate whether CC can directly inhibit the activation of TLR2 induced by chitin released from *Demodex*, and indirectly inhibit the activation of KLK5 through the inhibition of TLR2 activation. Our findings offer a hypothesis that may explain the therapeutic potential of CC and could have broad implications for the treatment of rosacea.

## 2. Results

### 2.1. Chemical Composition Analysis of Coptis Chinensis Franch

The components identified in the CC water extracts (CCE) are alkaloids and the dominant component was berberine. The CCE contained 15.8% berberine, 4.8% coptisine, and 4.8% palmatine. Chromatograms of typical extracts and mixed standards are shown in Figure 1A,B. The results of the quantitative analysis are summarized in Figure 1C. Peak identification was achieved by comparing the retention times to the UV spectra obtained for individual standards (Appendix A).

### 2.2. CCE Decrease KLK5 and Cathelicidin Expression in Epidermal Keratinocytes In Vitro

To confirm the expression of KLK5 and cathelicidin in human epidermal keratinocytes, HEKn, we examined the kinetics of KLK5 and cathelicidin expression induced by VD_3_. KLK5 mRNA expression persistently increased for 48 h, and cahtelicidin mRNA increased for 24 h after treatment with VD_3_, which gradually decreased (Figure 2A,B). KLK5 protein expression persistently increased for 96 h, while cathelicidin protein expression increased for 48 h after VD_3_ treatment, and thereafter remained constant (Figure 2C,D). Based on these results, the effect of CCE on the expression of KLK5 and cathelicidin was confirmed in HEKn induced by VD_3_. CCE downregulated the protein levels of KLK5 and cathelicidin induced by VD_3_ (Figure 2E,F). Furthermore, after treatment with CCE, the mRNA levels of KLK5 and cathelicidin antimicrobial peptides (CAMP) decreased, showing a dose-dependent pattern (Figure 2G,H).

### 2.3. CCE Decrease Cutaneous Trypsin-Like Serine Protease (KLK5) Activity

We further investigated whether CCE could attenuate the KLK5 protease activity in enzymatic reaction using a trypsin-like serine protease-specific fluorogenic peptide substrate, Boc-Val-Pro-Arg-7-amido-4-methylcoumarin hydrochloride (Boc-V-P-R-AMC). Leupeptin is known as a protease inhibitor that inhibits serine protease, such as trypsin, plasmin, and kallikrein [25]. Since KLK5 is a stratum corneum tryptic enzyme (SCTE), it was used as a positive control. Figure 3A shows that recombinant human KLK5 (rhKLK5) cleaved the substrate to generate fluorogenic product, and CCE significantly inhibited the enzymatic activity, compared with the control, in fluorogenic substrate kinetic assays. In confirming the inhibition of KLK5 enzymatic activity, the efficacy in HEKn was also confirmed. Whether CCE can have an effect on the activation of KLK5 induced by VD_3_ in HEKn was investigated. The results confirmed that CCE effectively inhibits the secreted KLK5 protease activity induced by VD_3_ (Figure 3B).

### 2.4. CCE Inhibit the Proteolytic Cleavage of Cathelicidin by Inhibiting KLK5 Expression and Protease Activity

Cathelicidin is stored as an inactive pro-form in intracellular granules, and is activated by proteolytic processing by KLK5 after extracellular secretion. Serine protease, KLK5 directly activates the inactive pro-cathelicidin into bioactive fragment LL-37 through a proteolytic process [2]. Therefore, we evaluated the ability of CCE to attenuate the processing of pro-cathelicidin into LL-37 induced by VD_3_ in HEKn. HEKn was pretreated with CCE, and then incubated with VD_3_ for 48 h. Western blot analysis for LL-37 was performed on HEKn culture medium. Cathelicidin expression was very weakly detected under vehicle treated condition (Figure 4, lane 1), whereas VD_3_-treated HEKn culture medium detected expressed cathelicidin and processed LL-37 (Figure 4, lanes 2–6). In HEKn treated with VD_3_, it was confirmed that the cathelicidin precursor was cleaved to LL-37 by activated KLK5 (Figure 4, lane 2), indicating that proteolytic processing was inhibited by CCE and its specific serine protease inhibitor, leupeptin (Figure 4, lanes 3–6). These results suggest that CCE suppressed the proteolytic processing of cathelicidin secreted from HEKn, by blocking the enzymatic activity of KLK5.

### 2.5. CCE Decrease the Expression of Pro-Inflammatory Cytokines Induced by VD_3_

LL-37 is an important effector molecule of innate immunity in the inflammatory skin disorder, rosacea. In an animal model, intradermal injection of LL-37 induced an inflammatory response with rosacea-like features [6]. LL-37 stimulates the secretion of immune mediators by keratinocytes, epithelial cells that include chemokines and cytokines. To evaluate the role of LL-37 as a mediator of inflammation, we stimulated HEKn and THP-1. In our experimental conditions, LL-37 in HEKn hardly induced the expression of pro-inflammatory cytokines, while in THP-1, it effectively induced the expression (Appendix A). In addition, VD_3_ did not significantly induce the expression of cathelicidin LL-37 directly in THP-1, compared with HEKn (Appendix A). Based on these results, inflammatory rosacea model was constructed through a transwell co-culture system of upper inserts_HEKn and bottom_THP-1. This in vitro co-culture system is an inflammatory rosacea model that induces inflammatory responses by molecular interactions between epidermal keratinocytes and immune cells and was used to evaluate the effect of CCE on inflammatory rosacea induced by VD_3_. No significant change by treatment with VD_3_ was observed in the expression of pro-inflammatory cytokines in single culture of either HEKn or THP-1, whereas HEKn co-cultured with THP-1 induced these levels, relative to the single culture (Figure 5A,B). Pro-inflammatory cytokines were increased upon exposure to VD_3_. The treatment of CCE with VD_3_ prevented the decrease of VD_3_-induced pro-inflammatory cytokines at both mRNA and protein level (Figure 5A,B). These results indicate that VD_3_ secretes LL-37 in HEKn, which induces the expression of pro-inflammatory cytokines in THP-1; and CCE was able to suppress the expression of pro-inflammatory cytokines by inhibiting the expression and processing of LL-37.

### 2.6. Chitin-Induced Inflammatory Responses Are Inhibited by CCE

*Demodex* mites may contribute to the pathogenesis of rosacea by stimulating the innate immune system [13,14]. The mites’ chitinous exoskeletons act as pathogen-associated molecular patterns (PAMPs), with the chitin released from *Demodex* mites potentially provoking inflammatory response through a TLR2 signaling pathway [15,26]. Since chitin directly binds to the TLR2 ectodomain, interference with chitin-TLR2 interaction suppresses inflammatory signaling in response to chitin [26]. To determine whether chitin regulates inflammatory response, we evaluated the effects of chitin fragments on macrophage inflammatory response. Chitin elicited NF-κB activation in TLR2/NF-κB-HEK293T cells, which signal transduction is inhibited by CCE (Figure 6A). Comparable results were obtained with CCE, CU-CPT22 (TLR1/TLR2 antagonist) on chitin-induced TLR2 activation (Figure 6A). These results suggest that chitin may mediate an immune response through binding to TLR2/TLR1 heterodimer, while CCE significantly inhibited its signal transduction (Figure 6B). In addition, chitin induced TLR2 expression, which was inhibited by CCE (Figure 6F). Furthermore, chitin-induced TLR2 activation stimulates the expression of KLK5 in HEKn under high calcium (1.5 mM) condition (Figure 6G). An increase in KLK5 expression induced by chitin was decreased by CCE, and CU-CPT22, an antagonist of TLR2/1 (Figure 6G). In HEKn culture, the expression of KLK5 was increased under high-concentration calcium (1.5 mM) condition; and when TLR2 was activated by chitin, KLK5 expression was accelerated (Figure 6G). These results imply that the inhibition of TLR2 activation from the agonist can act as a target for inhibiting the expression of KLK5 in rosacea. Activated TLR2 signaling results in the production of inflammatory mediators, which lead to the manifestation of inflammatory responses. 

We next investigated whether CCE could attenuate the expression of pro-inflammatory cytokines mediated by chitin-induced TLR2 signaling pathway. The ability of CCE to inhibit TLR2 signal transduction was reflected in the inhibition of pro-inflammatory cytokines expression (Figure 6C–E). These results suggest that CCE showed the effect of significantly inhibiting the expression of pro-inflammatory cytokines induced by chitin, which is through the inhibition of TLR2 activation and expression.

### 2.7. LL-37-Induced Vascular Endothelial Cell Proliferation Was Inhibited by CCE

Erythematous rosacea is characterized by transient flushing and persistent facial erythema. Vascular dysregulation has been considered to be primarily implicated in the etiology of erythematous rosacea, which causes angiogenesis, leading to the formation of new blood vessels [27,28]. To evaluate the effect of CCE on angiogenesis, we conducted proliferation assay in human microvascular endothelial cells (HMEC-1) induced by LL-37. HMEC-1 proliferation was evaluated by water-soluble tetrazolium salt (WST) and Ki-67, a hallmark of cell proliferation. Glutamine, known as endothelial growth factor [29], was used as a control to compare the proliferative effect of LL-37 on HMEC-1. Figure 7A shows that LL-37 induces HMEC-1 proliferation comparable to the positive control glutamine, which is inhibited by CCE in a dose-dependent manner, with 25% inhibition at a CCE concentration of 20 µg/mL. In addition, we evaluated the immunofluorescence staining of Ki-67 in HMEC-1, by flow cytometry. Similar to the results from WST assay, CCE significantly decreased LL-37-induced Ki-67 expression (Figure 7B). The Ki-67 positive cells increased by 20% compared to the untreated control, while CCE treatment suppressed them by 15.5% (Figure 7B). These results suggest that by effectively inhibiting the proliferation of vascular endothelial cells, CCE is expected to be applicable to the treatment of vascular rosacea caused by vascular abnormalities.

## 3. Discussion

Skin homeostasis is maintained by the renewal and differentiation of keratinocytes. The stratum corneum is composed of terminally differentiated keratinocytes called corneocytes, and acts as a physical barrier. During homeostasis, stratum corneum layers continuously shed by desquamation [30,31]. In this process, the stratum corneum tryptic enzyme (SCTE, KLK5) plays a role in the degradation of corneodesmosomes, an adhesion molecule of corneocytes [32]. In healthy skin, the activity of KLK5 is constantly regulated to maintain skin homeostasis as desquamation is performed. If the activity of KLK5 is not regulated, immunoregulatory abnormality occurs, due to damage to the skin barrier function, which leads to skin diseases [6,33,34]. A typical disease is rosacea, in which the skin antimicrobial peptide cathelicidin, cleaved with LL-37 by the activation of excessive KLK5, increases the expression of inflammatory mediators, and causes erythema. Mechanisms that regulate this network of KLK5 in the epidermis are important in regulating epidermal homeostasis through normal physiological interactions, and understanding these is an important research area, as changes in proteolytic balance can lead to skin inflammation and erythema. CC is widely used as an anti-inflammatory agent, and recent researches have studied a variety of bioactive mechanisms [19,20,21,22,23,24]. However, their effects on rosacea and their mechanisms of action have not yet been elucidated. In this study, we propose the effects of CC on the inflammation and erythema of rosacea, and its potential for the development of therapeutic agents.

Although the exact cause of rosacea is unknown, it is likely to occur due to a combination of genetic and environmental factors. Factors that may trigger rosacea include sunlight, *Demodex* mites, stress, and heat, of which sun exposure is the most common trigger. UV radiation is the major source for the synthesis of vitamin D, providing the promotion of bone and musculoskeletal health, and reducing the risk of a number of cancers [35]. However, in rosacea patients, VD_3_ increases skin serine protease activity and cathelicidin expression, resulting in inflammation and erythema. VD_3_ induces the expression of KLK5 and cathelicidin, and when cultured in HEKn for 96 h, showed almost the same level of expression (Figure 2A,B). CCE showed the effect of inhibiting both the expression of cathelicidin and KLK5 induced by VD_3_. Since cathelicidin expression is upregulated through binding of the nuclear receptor VDR to the VDRE site of the cathelicidin promoter, CCE may have a direct effect on the transcriptional phase of cathelicidin induced by VD_3_. The human KLK10 promoter has been reported to have retinoid X response element, but KLK5 promoter has not been clearly reported to have an RXR/VDRE binding site [36,37]. Further studies of the KLK5 promoter binding site will provide better understanding of the expression of KLK5 for transcriptional regulation mechanisms.

Patient with rosacea is characterized by the increased expression of KLK. Since pro-KLK5 has the ability to activate itself [38], inhibiting their protease activity is one of the main targets for rosacea treatment. CCE not only showed the effect of inhibiting the expression of KLK5, but also showed the ability to inhibit KLK5 protease activity in enzyme kinetic assay (Figure 2 and Figure 3). CCE also inhibited enzymatic processing cathelicidin through the inhibition of KLK5 protease activity in HEKn (Figure 4). In HEKn experiments, pro-cathelicidin processing is generated by extracellularly secreted KLK5, so the downregulated expression of KLK5 and cathelicidin by CCE may result in inhibition of the processing of LL-37. These results indicate that CCE exhibits an inhibitory effect on KLK5 and cathelicidin expression and KLK5 protease activity, and thus it is likely to be effectively applicable to erythema and inflammation in rosacea induced by processed LL-37.

CCE is mainly composed of berberine, coptisine and palmatine (Figure 1). CCE has been shown to effectively inhibit KLK5 activity at 20 ppm (Figure 3), and its effective concentration contains 9.23 µM berberine, 2.85 µM coptisine, and 2.8 µM palmatine. The major constituents, berberine and coptisine, showed significant inhibitory effect on KLK5 activity, whereas palmatine did not show statistical significance (Appendix A). This result confirms that these alkaloids are active ingredients of CCE that inhibit KLK5 activity. Further active component identification and bioavailability studies will help develop CCE as a treatment for rosacea skin disease.

*Demodex* mites are the most common permanent ectoparasites in human and inhabit the hair follicles or sebaceous glands [39]. *Demodex* mites, which are also present in healthy human skin, do not exhibit specific skin symptoms, while increased density of *Demodex* has various clinical manifestations. In rosacea patients, *Demodex* density appears higher than in healthy skin, which higher density activates downstream signaling pathway of TLR2, and increases the expression of inflammatory mediators. At this step, CCE has the ability to inhibit TLR2 expression, and also to act on the downstream signaling of TLR2, effectively suppressing the expression of inflammatory mediators. In addition, activation of TLR2 increases the influx of calcium into cells and induces KLK5 expression, which induces LL-37 activation, thus accelerating the inflammatory responses5. Therefore, it is an important target in regulating the skin immune response by inhibiting the activation of TLR2 signaling in rosacea. Since CCE effectively acts on the expression of TLR2 and its downstream signaling pathway, it is expected that it offers potential as an anti-inflammatory material for rosacea.

Another major symptom in patients with rosacea is persistent facial erythema, which is caused by the angiogenesis and vasodilation of vascular endothelial cells. In the molecular and clinical characteristics of early-stage rosacea, vasodilation is primarily observed, but not angiogenesis [40]. Most of the currently developed agents for erythematous rosacea are alpha adrenergic receptor agonists, anti-vasodilators that induce vascular smooth muscle contraction [41,42,43]. These agents are widely used as treatments for patients with erythematous rosacea, due to their immediate effect; however, because it is not a treatment for the underlying cause, its effects are limited, so it is difficult to show medicinal efficacy in patients with complex symptoms. In this study, CCE was observed to be effective not only for suppressing inflammation, but also for inhibiting vascular endothelial cell growth, so it is expected to be developed as a therapeutic agent having the complex effect of improving inflammation and erythema (Figure 8).

In this study, we demonstrated the effect and mechanism of CCE on rosacea and found that it was through regulation of the inflammatory responses and angiogenesis. Our findings are a preliminary stage to demonstrate the efficacy of CCE in immunomodulation and improvement of erythema in rosacea. This study showed the potential of CCE as a therapeutic agent for rosacea, and further studies will be needed to clarify its clinical efficacy.

## 4. Materials and Methods

### 4.1. Cell Culture

Human epidermal keratinocytes (HEKn, Gibco, Waltham, MA, USA) were maintained in EpiLife medium (Gibco) supplemented with Human Keratinocyte Growth Supplement (HKGS, Gibco) at 37 °C, under 5% CO_2_. Human monocytes, THP-1 were maintained in RPMI 1640 medium (WELGENE Inc., Gyeongsan, Korea), containing 10% FBS and 1% penicillin/streptomycin (Gibco) at 37 °C, under 5% CO_2_. Human endothelial microvascular cells (HMEC-1, CRL-3243™, ATCC^®^ Manassas, VA, USA) were maintained in MCDB 131 (Gibco), containing 10 ng/mL EGF (Invitrogen, Carlsbad, CA, USA), 1 µg/mL hydrocortisone (Sigma-Aldrich, St. Louis, MO, USA), 10 mM glutamine (Gibco), and 10% FBS at 37 °C, under 5% CO_2_.

### 4.2. Preparation of Coptis Chinensis Franch Extract (CCE)

The root of *Coptis chinensis* Franch (CC) was obtained from the Samhong Herb-Medicine Co. (Seoul, South Korea). Water extract of CC was prepared by reflux extraction in purified water at 90–95 °C for 3 h. The extracts were filtered through filter paper. After spray drying, a perfectly dried extract of CC was obtained. The obtained extract was dissolved in distilled water for further experiments. 

### 4.3. Cell Culture

The root of *Coptis chinensis* Franch (CC) was obtained from the Samhong Herb-Medicine Co. (Seoul, Korea). Water extract of CC was prepared by reflux extraction in purified water at 90–95 °C for 3 h. The extracts were filtered through filter paper. After spray drying, a perfectly dried extract of CC was obtained. The obtained extract was dissolved in distilled water for further experiments.

### 4.4. High-Performance Liquid Chromatography (HPLC)

The aqueous extract of CC was quantitatively analyzed by HPLC [44,45]. The HPLC system used in this study was a Waters 2695 (Milford, MA, USA), equipped with a Waters 2996 Photodiode Array (PDA Detector). The Empower 2 software was used to control the analytical system and perform the data collection and processing. For HPLC-PDA was performed on a Luna C_18_(2) (4.6 × 250 mm, 5 μm) column reversed-phase column protected by a C18 guard column from Phenomenex, Inc. (Torrance, CA, USA). The sample injection volume was 10 µL. The signal was monitored at 350 nm. The elution system used for the HPLC-PDA assay was a binary high-pressure gradient elution system with mobile phase A (0.1% trifluoroacetic acid, TFA in H_2_O) and mobile phase B (acetonitrile). Elution gradient: 20% organic phase B, hold for 5 min; from 20 to 40% organic phase B in 30 min (linear gradient), hold for 10 min; from 40 to 100% organic phase B in 10 min (linear gradient), hold for 10 min; then back to the starting condition in 1 min and re-equilibration for 9 min. The flow rate was 1.0 mL/min. Each analysis required 70 min, including the re-equilibration time.

#### 4.4.1. Chemical Standards and Solvents

Coptisine and palmatine standard were obtained from ChemFaces (Wuhan, Hubei, China). Berberine standard was purchased from Sigma–Aldrich. HPLC grade acetonitrile (≥99.9%) and methanol were purchased from J.T. Baker (Phillipsburg, NJ, USA) and TFA (≥99.0%) were purchased from Sigma-Aldrich.

#### 4.4.2. Preparation of Standard and CCE Solution

The Coptisine, palmatine and berberine standard were dissolved in methanol (0.5–100.0 mg/L) with an appropriate sonication, and CCE was dissolved in methanol (1.0 g/L) with an appropriate sonication. The standard and test solutions were filtered through a 0.22 μm syringe filter (SmartPor-II with universal hydrophilic polytetrafluoroethylene membrane, Sigma-Aldrich) prior to performance of the HPLC injection.

#### 4.4.3. Validation of the Method

Method validation was conducted based on international conference on harmonisation (ICH) guidelines [46]. For selectivity tests, this method has shown that coptisine, palmatine, and berberine were well separated. The three standard analytical parameters of the method of chromatographic analysis developed have been calculated: Coptisine was linear range (0.5–100.0 mg/L, r2 = 0.998), detection limit (0.3 mg/L), quantification limit (0.9 mg/L). Palmatine and berberine were linear range (0.5–100.0 mg/L, r2 = 0.998), detection limit (0.2 mg/L), quantification limit (0.6 mg/L). Using this method, CCE have been analyzed and the concentration of three materials in these has been determined.

### 4.5. TLR2/NF-κB/SEAP Activity

HEK293 cells that stably co-express the human TLR2/1 or TLR2/6 and an NF-κB-inducible secreted alkaline phosphatase (SEAP) reporter gene were purchased from InvivoGen (HEK-Blue™ hTLR2, San Diego, CA, USA). HEK-hTLR2 cells were maintained in DMEM containing 10% FBS, 100 µg/mL Normocin™ (InvivoGen), and 1 × HEK-Blue™ Selection (InvivoGen) at 37 °C, under 5% CO_2_. HEK-hTLR2 cells were seeded in 96-well plates at 5 × 10^4^ cells per well in 200 µL HEK-Blue™ Detection (InvivoGen). The cells were pretreated with CCE and TLR2 antagonist, CPT-CU22 (Sigma-Aldrich) for 1 h, followed by stimulation with 100 µg/mL chitin (Sigma-Aldrich) for 12 h. TLR2/NF-κB-induced SEAP activity was determined by reading the OD at 655 nm.

### 4.6. TLR2 Expression Analysis

HEKn were seeded in triplicate wells of 6-well plate at 5 × 10^4^ cells/well, and incubated overnight. The cells were pretreated with CCE at 5, 10, and 20 µg/mL for 1 h, and then further incubated with chitin (100 µg/mL) for 48 h. Cells were harvested, cDNA was synthesized, and TLR2 mRNA was then measured by quantitative real-time PCR (7500 Real-Time PCR Instrument System, Applied Biosystems, Foster City, CA, USA).

### 4.7. HMEC-1 Proliferation Assay

#### 4.7.1. WST Assay

HMEC-1 proliferation was measured using the water-soluble tetrazolium salt (WST, EZ-Cytox, DoGenBio, Seoul, Korea). Cells were plated in triplicate wells of 24-well plate, and incubated overnight. The cells were pretreated with CCE at (5, 10, and 20) µg/mL or 1 mM L-glutamine (Gibco), and then treated with 5 µg/mL LL-37 (Carbosynth, Berkshire, UK) under supplement-free condition. After 72 h incubation, Cells were treated with WST, and further incubated for 2 h. Absorbance was measured at 450 nm wavelength using microplate spectrophotometry. As a positive control, cells were cultured in 1 mM L-glutamine.

#### 4.7.2. Ki-67 Assay

Cells were plated in triplicate wells of 6-well plate, and incubated overnight. The cells were pretreated with 20 µg/mL CCE, followed by treatment with 5 µg/mL LL-37 (Carbosynth) under supplement-free MCDB 131 medium condition. In the next 48 h of treatment, cells were harvested, and stained with monoclonal Ki-67-FITC antibody (eBioscience, Carlsbad, CA, USA) and Rat IgG2a kappa Isotype Control-FITC (eBioscience). Ki-67 expression was evaluated by flow cytometry assay.

### 4.8. Preparation of Chitin Particles

Chitin fragments were generated as previously described [47,48]. Briefly, chitin powder (Sigma-Aldrich) was suspended in 1× PBS, and sonicated at 25% amplitude three times for 20 min with a Vibra-Cell™ ultrasonic sonicator (Sonics & Materials, Inc., Newtown, CT, USA). Suspended particles were filtered with (100, 70, and 40) µm cell strainer (SPL Life Sciences, Pocheon, Korea), and (40–70) µm size particles were then lyophilized.

### 4.9. KLK5 Protease Activity

KLK5 activity was assayed by recombinant human KLK5 proteins (rhKLK5, R&D systems, Minneapolis, MN, USA) with NaH_2_PO_4_ buffer (pH 8.0) at room temperature. KLK5 protease activity was measured by its ability to cleave the fluorogenic peptide substrate, Boc-V-P-R-AMC (R&D systems) in kinetic assay. Briefly, recombinant human KLK5 was preincubated with CCE at 5, 10, or 20 µg/mL or 1 µM leupeptin, a serine protease inhibitor for 10 min, followed by the addition of substrate. Relative fluorescence unit (RFU) was measured at Ex 380 nm/Em 460 nm in kinetic mode for 5 min. Relative KLK5 activity was calculated based on the formula: (*V*max of test sample—*V*min of test sample)/(*V*max of vehicle—*V*min of vehicle) × 100.

### 4.10. Cathelicidin Cleavage Assay

HEKn were plated in 12-well plate at 1 × 10^5^ cells/well and incubated until 90–100% confluency was reached. HEKn was pretreated with CCE at 5, 10, or 20 µg/mL or leupeptin (final 1 µM) for 2 h, followed by treatment with VD_3_ at 37 °C for 48 h. Culture medium was collected, and lyophilized. The sample was dissolved with the same volume of deionized water; and after running the Tricine-SDS-PAGE system (Koma Biotech, Seoul, Korea), western blot analysis was performed with anti-LL-37 antibody (Novus Biologicals, Centennial, CO, USA).

### 4.11. Co-Culture Conditions

For co-culture experiments, we used Millicell Cell culture insert (Millipore, Billerica, MA, USA), according to the manufacturer’s instructions. Briefly, HEKn cells were seeded on 0.4 μm pore transwell inserts at a density of 1 × 10^3^ cells, and maintained for 48 h. THP-1 cells were seeded in 12-well plates at 5 × 10^5^ cells/well, and differentiated into macrophages using 100 nM phorbol 12-myristate 13-acetate, PMA (Sigma-Aldrich) for 24 h. The culture media of both cells were replaced with fresh DMEM containing 1% FBS and 1% penicillin/streptomycin (Gibco). Subsequently, transwell inserts were placed in the upper compartment of THP-1 seeded 12-well plates. Upper inserts were pretreated with CCE for 1 h, followed by treatment with VD_3_. After 72 h incubation, secreted pro-inflammatory cytokines were determined by ELISA.

### 4.12. Total RNA Extraction, cDNA Synthesis, and Quantitative PCR

Total RNA extraction was performed using RNeasy kit (Qiagen, Hilden, Germany), and cDNA was synthesized using an AccuPower^®^ CycleScript RT PreMix (Bioneer, Daejeon, Korea), according to the manufacturer’s instructions. The KLK5, CAMP, IL-1β, IL-8, TNF-α, and TLR2 mRNA were measured by real-time quantitative PCR. The primer sequences were as follows: KLK5: forward 5′-CGTCCCACTAAAGATGTCAGACC-3′, reverse 5′-TCAAGCACTGGAGGACCTTAGG-3′; CAMP: forward 5′-GACACAGCAGTCACCAGAGGAT-3′, reverse 5′-TCACAACTGATGTCAAAGGAGCC-3′; IL-1β: forward 5′-CCACAGACCTTCCAGGAGAATG-3′, reverse 5′-GTGCAGTTCAGTGATCGT ACAGG-3′; IL-8: forward 5′-GAGAGTGATTGAGAGTGGACCAC-3′, reverse 5′-CACAACCCTCTG CACCCAGTTT-3′; TNF-α: forward 5′-CTCTTCTGCCTGCTGCACTTTG -3′, reverse 5′-ATGGGCTACAGGCTTGTCACTC -3′; TLR2: forward 5′-CTTCACTCAGGAGCAGC AAGCA-3′, reverse 5′-ACACCAGTGCTGTCCTGTGACA-3′; GAPDH: forward 5′-TGCACCACCAAC TGCTTAGC-3′, reverse 5′-GGCATGGACTGTGGTCATGAG-3′. All mRNA data were normalized to GAPDH expression.

### 4.13. Statistical Analysis

Differences between the control and treatment group were analyzed by Student’s t-test. *p* < 0.05 was considered statistically significant.

## Figures and Tables

**Figure 1 molecules-25-05556-f001:**
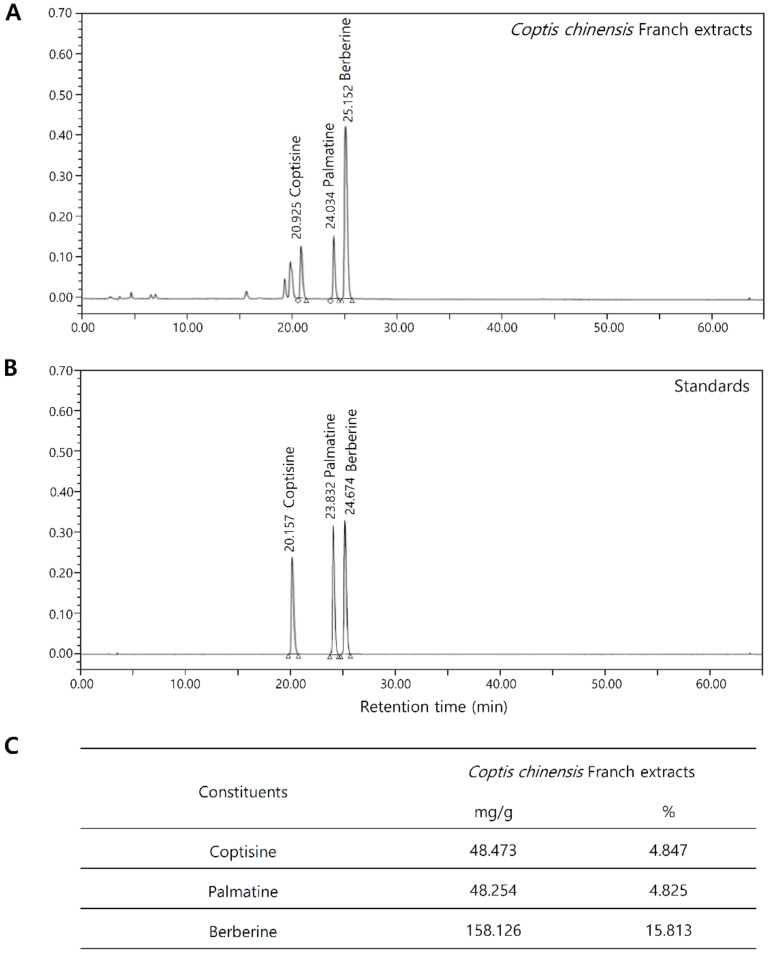
Composition of *Coptis chinensis* Franch extract (CCE). (**A**,**B**) Phytochemical analysis of CCE. (**C**) Quantitative analysis of the phytochemical activity of CCE.

**Figure 2 molecules-25-05556-f002:**
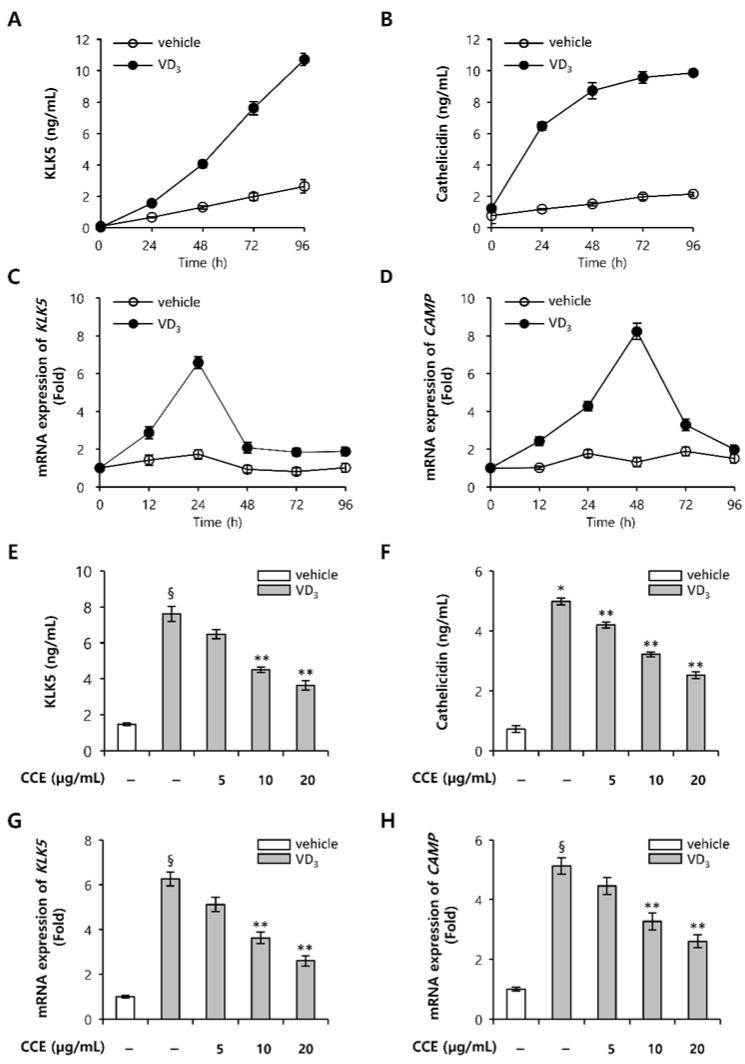
CCE inhibits KLK5 and cathelicidin expression in human epidermal keratinocyte induced by VD_3_. HEKn (passage 3) at 90% confluence was stimulated with VD_3_ for 12, 24, 48, 72, or 96 h. (**A**,**B**) The expression of KLK5 and cathelicidin protein was measured by ELISA, while (**C**,**D**) KLK5 and cathelicidin mRNA were measured by quantitative real-time PCR. HEKn (passage 3) at 90% confluence was pre-incubated with 5, 10, or 20 µg/mL CCE for 2 h, and subsequently treated with 200 nM VD_3_ for 96 h. (**E**,**F**) The expression of KLK5 and cathelicidin protein was measured by ELISA. (**G**,**H**) KLK5 and cathelicidin mRNA were measured by quantitative real-time PCR. The results are mean ± standard deviation (SD) (*n* = 3). * *p* < 0.001 vs. vehicle control; § *p* < 0.01 vs. vehicle control; ** *p* < 0.01 vs. vehicle control; DMSO vehicle control (vehicle).

**Figure 3 molecules-25-05556-f003:**
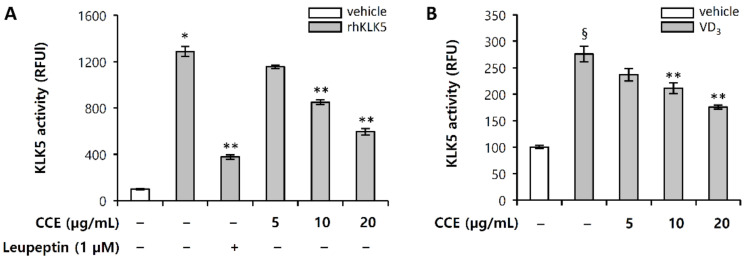
CCE inhibit KLK5 protease activity. KLK5 activity was measure in relative fluorescence units (RFU), using fluorogenic peptide substrate sensitive to KLK5. (**A**) Recombinant human KLK5 (20 ng) was preincubated with 5, 10, or 20 µg/mL CCE and 5 µM leupeptin, a serine protease inhibitor, for 15 min, and then further incubated with 100 µM Boc-V-P-R-AMC, fluorogenic peptide substrate, for 5 min. (**B**) HEKn (passage 3) at 80% confluence was pretreated with 5, 10, or 20 µg/mL CCE for 1 h, and subsequently treated with 200 nM VD_3_ for 72 h. HEKn culture media was collected, and lyophilized. The sample was measured for KLK5 activity using 100 µM Boc-V-P-R-AMC, fluorogenic peptide substrate. The results are mean ± standard deviation (SD) (*n* = 3). * *p* < 0.001 vs. vehicle control; ** *p* < 0.01 vs. rhKLK5-treated control; § *p* < 0.01 vs. vehicle control; *p* < 0.01 vs. VD_3_-treated control; DMSO vehicle control (vehicle).

**Figure 4 molecules-25-05556-f004:**
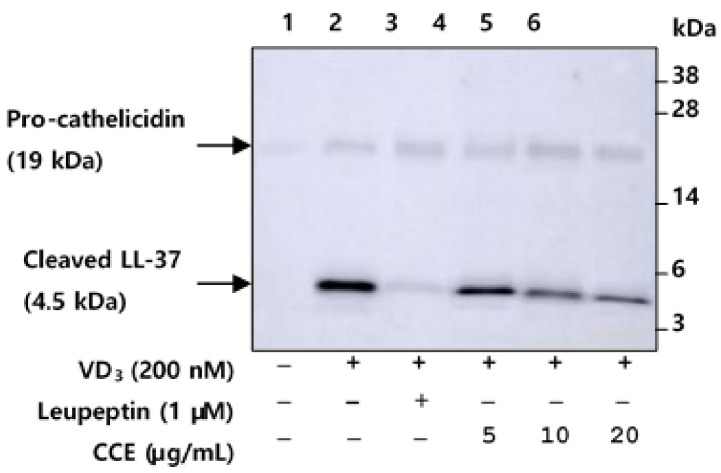
CCE inhibits the proteolytic cleavage of cathelicidin by inhibiting KLK5 activation. HEKn (passage 3) at 90% confluence was pretreated with 5, 10, or 20 µg/mL CCE and 5 µM leupeptin, a serine protease inhibitor, for 1 h, and subsequently treated with 200 nM VD_3_ for 48 h. HEKn culture media was collected, and analyzed by Western blot analysis with anti-LL-37 antibody. Lane 1, DMSO vehicle control; lane 2, HEKn + VD_3_; lane 3, HEKn + VD_3_ + leuepetin; lanes 4–6, HEKn + VD_3_ + 5, 10, or 20 µg/mL CCE. Full-length blots are presented in Appendix A.

**Figure 5 molecules-25-05556-f005:**
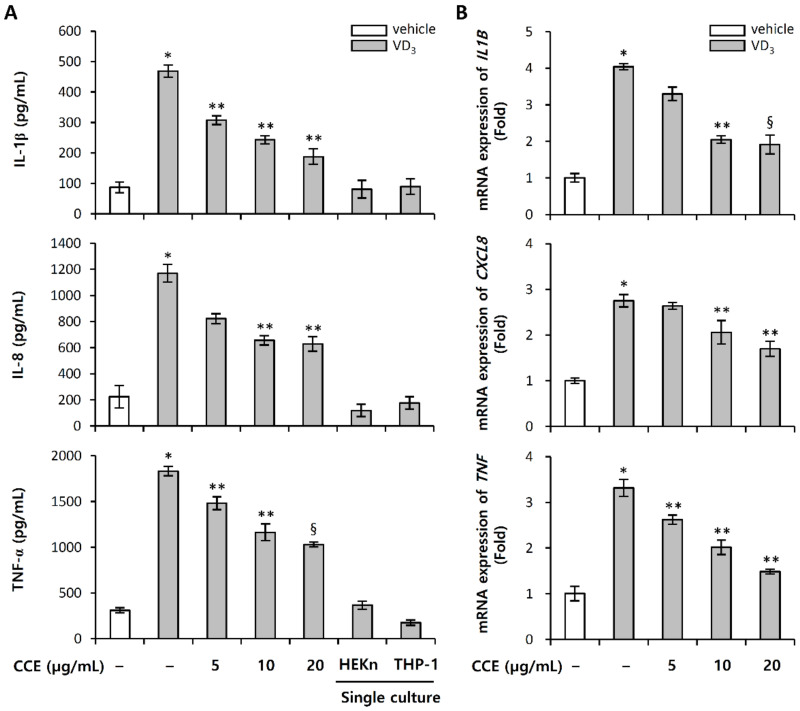
CCE inhibits the expression of pro-inflammatory cytokines induced by VD_3_ in HEKn/THP-1 co-culture system. HEKn (passage 3) was seeded in upper insert chambers, and THP-1 was seeded in lower chamber of two-compartment transwell system separately for 24 h. Upper insert chamber was pretreated with 5, 10, or 20 µg/mL CCE for 1 h, and subsequently treated with 200 nM VD_3_. (**A**) After 72 h incubation, secreted pro-inflammatory cytokines (IL-1β, IL-8, TNF-α) were determined by ELISA. Pro-inflammatory cytokines (IL-1β, IL-8, TNF-α) mRNA were measured by quantitative real-time PCR. (**B**) mRNA expression was normalized as the relative expression of GAPDH mRNA, and all data are presented as fold induction against control. The results are mean ± standard deviation (SD) (*n* = 3). * *p* < 0.001 vs. vehicle control; ** *p* < 0.01 vs. VD_3_-treated control; § *p* < 0.05 vs. VD_3_-treated control; DMSO vehicle control (vehicle).

**Figure 6 molecules-25-05556-f006:**
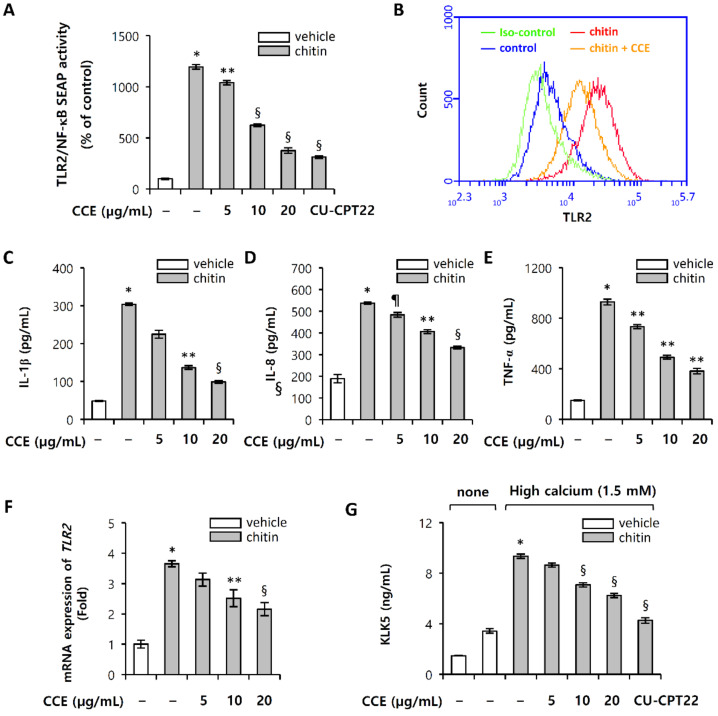
Chitin-induced TLR2 signaling pathways are regulated by CCE. (**A**) HEK-hTLR2 cells were cultured in DMEM containing 10% FBS, 100 µg/mL Normocin™, and 1× HEK-Blue™ Selection. HEK-hTLR2 cells were pretreated with (5, 10, or 20) µg/mL CCE and CU-CPT22 (200 nM), a TLR2/1 antagonist, for 1 h, and subsequently treated with 100 µg/mL chitin for 12 h. TLR2/NF-κB 006B -induced SEAP activity was determined by reading the OD at 655 nm. (**B**) THP-1 was pretreated with 20 µg/mL CCE, and subsequently treated with 100 µg/mL chitin for 24 h. Cells were harvested, and stained with monoclonal TLR2-fluorescein isothiocyanate (FITC) antibody and mouse IgG2a kappa Isotype Control-FITC. TLR2 expression was analyzed by flow cytometry. (**C**–**E**) Pro-inflammatory cytokine protein was measured by ELISA. (**F**,**G**) HEKn (passage 4) was pretreated with 5, 10, or 20 µg/mL CCE and 200 nM CU-CPT22, a TLR2/1 antagonist, for 1 h, and subsequently treated with 100 µg/mL chitin for (48 or 72) h. TLR2 mRNA was measured by quantitative real-time PCR at 48 h. mRNA expression was normalized as the relative expression of GAPDH mRNA, and all data are presented as fold induction against control. The expression of KLK5 protein was measured by ELISA at 72 h. The results are mean ± standard deviation (SD) (*n* = 3). * *p* < 0.001 vs. vehicle control; § *p* < 0.001 vs. chitin-treated control; ** *p* < 0.01 vs. vehicle control; *p* < 0.05 vs. vehicle control; phosphate-buffered saline (PBS) vehicle control (vehicle).

**Figure 7 molecules-25-05556-f007:**
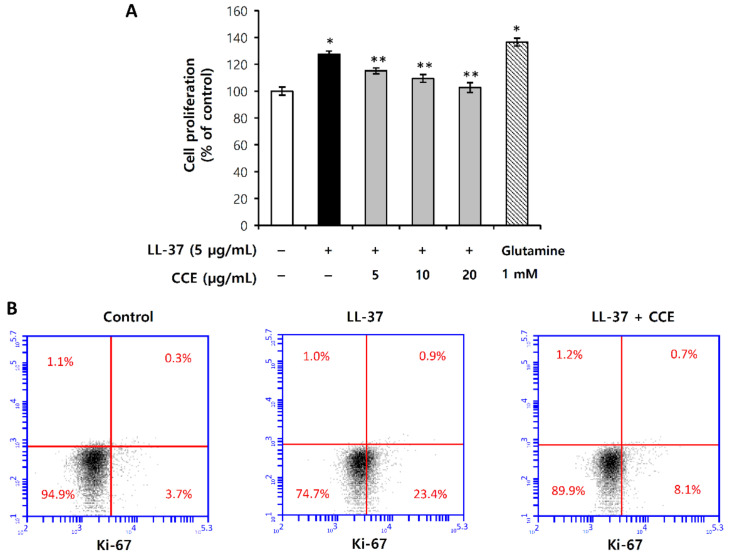
CCE inhibits HMEC-1 proliferation induced by LL-37. (**A**) HMEC-1 was pretreated with 5, 10, or 20 µg/mL CCE and 1 mM L-glutamine for 1 h, and then further incubated with 5 µg/mL LL-37 for 72 h. HMEC-1 proliferation was then determined by WST assay. (**B**) HMEC-1 was pretreated with 20 µg/mL CCE for 1 h, and then further incubated with 5 µg/mL LL-37 for 48 h. Cells were harvested, and stained with monoclonal Ki-67-FITC antibody and Rat IgG2a kappa Isotype Control-FITC. Ki-67 expression was analyzed by flow cytometry. The results are mean ± standard deviation (SD) (*n* = 3). * *p* < 0.01 vs. LL-37-untreated control; ** *p* < 0.05 vs. LL-37-treated control.

**Figure 8 molecules-25-05556-f008:**
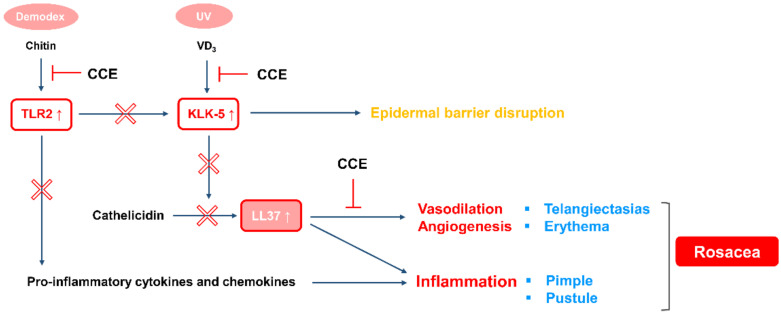
Mechanism of action of CCE in rosacea.

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
