# Peer review of "Coptis chinensis Franch Directly Inhibits Proteolytic Activation of Kallikrein 5 and Cathelicidin Associated with Rosacea in Epidermal Keratinocytes"

_molecules, 2020, doi:10.3390/molecules25235556_

Round 1
Reviewer 1 Report
The article submitted for review concerns a very interesting topic, namely the use of a medicinal plant Coptis chinensis Franch used in traditional folk medicine for the treatment of rosacea.
The paper is worth publishing In Molecules, however some small corrections should be made before publication.
Please provide full names for all abbreviations present in the manuscript as many are missing and this makes the text difficult to read.
Please improve the quality of the figures, because some of them are illegible (e.g. chromatograms in fig 1.).
Improvement in English language and style would also be appreciated.
The article submitted for review concerns a very interesting topic, namely the use of a medicinal plant Coptis chinensis Franch used in traditional folk medicine for the treatment of rosacea. The article has many strengths, such as a lot of biological research, e.g. analysis of the influence of CCE on the expression of KLK5 and cathelicidin in epidermal keratinocytes in vitro, assessment of the influence of CCE on the activity of cutaneous trypsin-like serine protease (KLK5), investigating CCE inhibition of proteolytic cleavage of cathelicidin by inhibiting KLK5 expression and protease activity, reducing the expression of VD3-induced pro-inflammatory cytokines, inhibition of chitin-induced inflammatory reactions, inhibition of LL-37-induced vascular endothelial cell proliferation, and therefore I recommended this work for publication.
The minor changes I have recommended relate to the improvement of the quality of the figures.
I also have some comments on the qualitative and quantitative analysis of the CCE extracts. Since they were conducted and mentioned in the article, they should be presented fairly.
The qualitative analysis should include the retention times of the peaks presented in the chromatograms 1A and 1B. In the case of coptisine, the peak of this compound in the chromatogram of the standards 1B is shifted towards the lower values of the retention time and there is a doubt whether the compound detected in the extract is actually coptisine (?). For this reason, the qualitative analysis should be supplemented by comparing the UV spectra of the compounds included in the extract with the UV spectra of the standards.
In the case of the quantitative analysis of the extract, it is described too briefly. Please include details of the quantification method and its validation in the supporting materials.
Reviewer 2 Report
Authors evaluated the effects of Coptis chinensis Franch, a medicinal herb in traditional oriental medicine, on rosacea in human epidermal keratinocytes.
Major revisions:
- Little is known about the Coptis chinensis Franch and more informations should be added. In particular Authors should specify the role of major active compounds in the observed effects.
- In the figures, untreated control should be better defined and reported.
-The Inflammatory rosacea model should be better described.
- Data concerning vascular endothelial cell proliferation seem not to show a really significant difference. Authors should make the information clearer.
Minor revisions
- A list of abbreviation should be provided.
- Many sentences should be rephrased and the manuscript should be revised by a native
speaker
